# Potential of Oral Nanoparticles Containing Cytokines as Intestinal Mucosal Immunostimulants in Pigs: A Pilot Study

**DOI:** 10.3390/ani12091075

**Published:** 2022-04-21

**Authors:** Adrià López-Cano, Alex Bach, Sergi López-Serrano, Virginia Aragon, Marta Blanch, Jose J. Pastor, Gemma Tedó, Sofia Morais, Elena Garcia-Fruitós, Anna Arís

**Affiliations:** 1Department of Ruminant Production, Institute of Agriculture and Food Research (IRTA), 08140 Caldes de Montbui, Spain; adrialopezcano@gmail.com (A.L.-C.); alex.bach@icrea.cat (A.B.); 2Institució Catalana de Recerca i Estudis Avançats (ICREA), 08010 Barcelona, Spain; 3IRTA, Centre de Recerca en Sanitat Animal (CReSA, IRTA-UAB), Campus de la Universitat Autònoma de Barcelona, 08193 Bellaterra, Spain; sergi.lopez@irta.cat (S.L.-S.); virginia.aragon@irta.cat (V.A.); 4OIE Collaborating Centre for the Research and Control of Emerging and Re-Emerging Swine Diseases in Europe (IRTA-CReSA), 08193 Bellaterra, Spain; 5Innovation Division, Lucta S.A., Edifici Eureka, UAB Research Park, 08193 Bellaterra, Spain; marta.blanch@lucta.com (M.B.); jose.pastor@lucta.com (J.J.P.); gemma.tedo@lucta.com (G.T.); sofia.morais@lucta.com (S.M.)

**Keywords:** immunostimulant, cytokines, nanoparticles, piglets, antimicrobial resistance

## Abstract

**Simple Summary:**

Antibiotics are essential compounds to cope with bacterial infections. However, their inadequate and excessive use has triggered the rapid arising of antimicrobial-resistant bacteria. In this scenario, immunostimulants, which are molecules that boost the immune system, open up a new approach to face this problem, enhancing treatment efficacy and preventing infections by immune system response. Cytokines are central effector molecules of the immune system, and their recombinant production and administration in animals could be an interesting immune modulation strategy. The aim of this study was the development of a highly stable nanoparticle of porcine cytokines to achieve the immunostimulation of intestinal mucosa in piglets. The outcomes of the present study prove this approach is able to stimulate swine intestinal cells and macrophages in vitro and tends to modulate inflammatory responses in vivo, although further studies are required to definitively evaluate their potential in animals.

**Abstract:**

Antimicrobial resistance is a global threat that is worryingly rising in the livestock sector. Among the proposed strategies, immunostimulant development appears an interesting approach to increase animal resilience at critical production points. The use of nanoparticles based on cytokine aggregates, called inclusion bodies (IBs), has been demonstrated as a new source of immunostimulants in aquaculture. Aiming to go a step further, the objective of this study was to produce cytokine nanoparticles using a food-grade microorganism and to test their applicability to stimulate intestinal mucosa in swine. Four cytokines (IL-1β, IL-6, IL-8, and TNF-α) involved in inflammatory response were produced recombinantly in *Lactococcus lactis* in the form of protein nanoparticles (IBs). They were able to stimulate inflammatory responses in a porcine enterocyte cell line (IPEC-J2) and alveolar macrophages, maintaining high stability at low pH and high temperature. In addition, an in vivo assay was conducted involving 20 piglets housed individually as a preliminary exploration of the potential effects of IL-1β nanoparticles in piglet intestinal mucosa after a 7 d oral administration. The treated animals tended to have greater levels of TNF-α in the blood, indicating that the tested dose of nanoparticles tended to generate an inflammatory response in the animals. Whether this response is sufficient to increase animal resilience needs further evaluation.

## 1. Introduction

Antibiotics are effective molecules to treat infectious diseases caused by bacteria. However, the overuse and misuse of these compounds have accelerated the emergence of antibiotic resistance, leading to the appearance of multiresistant bacteria that are easily transmitted between humans, animals, and the environment [1]. This has pushed the need to prioritize first-line antibiotics for human health, reduce their administration in livestock, and find new alternatives to the use of antibiotics to cope with resistant bacteria [2]. Antibiotic reduction in animal production mainly concerns preventive applications. In this context, new antimicrobial molecules are the focus of current research, but the use of immunostimulants to increase animal resilience at critical phases of animal production is a strategy that is also gaining interest [3]. For example, during the transport, housing, or weaning processes, livestock regularly suffer immunosuppression, metabolic dysregulation, and, as a result, the development of concomitant diseases [4]. In this scenario, immunostimulants hold the potential to boost the immune response to act faster and more efficiently when mitigating opportunistic pathogen infections. In addition, immunostimulant administration to the mother could be a powerful strategy to increase the quality of the colostrum and, therefore, newborn immune status [5,6].

Immunostimulants are substances (drugs or nutrients) that stimulate the complex and versatile biological network that composes the immune system. [7,8,9]. The development of immunostimulants in livestock is usually based on a nonspecific activity for the activation of the innate immunity of the animal. Moreover, immunostimulants can be used as vaccine adjuvants, improving vaccine efficacy by stimulating specific immunity.

The application of immunostimulants at the gastrointestinal level is encouraging because they can be administrated as a feed additive. Immunostimulants can target a wide variety of immune components involved in mucosal immunity and epithelial barrier function, comprising the microbiota and extending the effect systemically [10]. In this context, compounds based on flavonoids, essential oils, probiotics, or prebiotics have been deeply explored for livestock applications [11]. However, other molecules, such as lipopeptidases, lipopolysaccharide (LPS), flagellin, CpG nucleotides, and cytokines are also attractive immunostimulants that have been less investigated for animal production [12].

Nanoparticles have been used as therapeutic agents in the human medical field for some time now, though their application in veterinary medicine and animal production is still relatively new. Torrealba et al. [13] proposed the use of nanoparticles based on cytokine aggregates named inclusion bodies (IBs) as a new source of immunostimulants. They proved that the use of IBs, which are highly stable nanoparticles, produced in a single-step and cost-effective way showed outstanding in vivo immune protection in fish against an otherwise lethal *Pseudomonas aeruginosa* challenge [13]. However, the use of these cytokine-based nanoparticles has not been investigated in other species. Thus, we herein explore the concept of cytokine-based nanoparticles to boost innate immunity driven by swine intestinal mucosa as a possible proof of concept to further develop applications focused on increasing animal resilience during stressful production periods.

## 2. Materials and Methods

### 2.1. Bacterial and Culture Strains

*Lactococcus lactis subsp. cremoris* NZ9000 [14] was used for heterologous protein expression. *L. lactis* was grown in M17 medium supplemented with 0.5% glucose *v*/*v* (from now on, GM17), as previously described [15]. Immunoassays were performed using the intestinal porcine enterocyte cell line IPEC-J2 (DSMZ, German Collection of Microorganism and Cell Culture, Braunschweig, Germany) cultured at 37 °C and 5% CO_2_ in Dulbecco’s Modified Eagle’s Medium (DMEM) supplemented with 10% fetal bovine serum (FBS), 2 mM glutaMAX^TM^ (Thermo Scientific, Applied Biosystems, Gibco, Waltham, MA, USA), 0.5% *v*/*v* nystatin (Thermo Scientific, Applied Biosystems, Gibco, Waltham, MA, USA), insulin-transferrin-selenium (Thermo Scientific, Applied Biosystems, Gibco, Waltham, MA, USA), and penicillin-streptomycin (5.000 U/L, Thermo Scientific, Applied Biosystems, Gibco, Waltham, MA, USA). The alveolar macrophages used in the immunoassays were isolated from pig bronchoalveolar fluid, as previously described [16]. Briefly, after pig euthanasia, a bronchoalveolar lavage of the lungs was performed with 100 mL of sterile PBS supplemented with gentamicin at 70 μg/mL (Sigma-Aldrich, Madrid, Spain). Further, to collect the alveolar macrophages, the lavage fluids were centrifugated at 230× *g* for 15 min, and the cells were washed twice with DMEM containing gentamicin (50 μg/mL). Lastly, the alveolar macrophage concentration was adjusted to 1 × 10^7^ cells/mL, and aliquots were stored in DMEM with 10% dimethyl sulfoxide (DMSO) and 20% FBS.

### 2.2. Genetic Construct Design

Swine mature sequences of interleukin-1β (IL-1β) (115–267, Uniprot entry P26889), interleukin-6 (IL-6) (29–212, Uniprot entry P26893), interleukin-8 (IL-8) (26–104, Uniprot Entry 26894), tumor necrosis factor (TNF-α) (78–232, Uniprot Entry P23563) (using swine native sequences codon-optimized for expression in *L. lactis*), and green fluorescence protein (GFP) were chemically synthesized (GeneArt^®^, Lifetechnologies, Regensburg, Germany). All of them were cloned in pMA-T (Amp^R^) (GeneArt^®^, Regensburg, Germany) vector. Each sequence was flanked by NcoI and XbaI restriction enzyme sequences, allowing the subcloning of genes in the pNZ8148 (Cm^R^; MoBiTech, Richmond, VIC, Australia) vector suitable for the *L. lactis* expression system. All sequences also had a C-terminal 6-histidine tag for protein purification and quantification. Plasmids containing the sequences of interest were transformed into the electrocompetent *L. lactis* NZ9000 strain using a Gene Pulser (Bio-rad, Hercules, CA, USA) at 2500 V, 200 Ω, and 25 μF, as described by Cano-Garrido et al. [17].

### 2.3. Cytokine Nanoparticle Production and Purification

*Lactococcus lactis* NZ9000/pNZ8148 containing each cytokine gene was grown overnight (O/N) at 30 °C in GM17 supplemented with 5 μg/mL of chloramphenicol (Cm). Next, fresh GM17 (5 μg/mL Cm) was inoculated in the O/N cultures at an initial OD_600_ of 0.05. When the cultures reached OD_600_ = 0.4–0.6, they were induced with 12.5 ng/mL of nisin, starting the heterologous gene expression. The recombinant proteins were produced over 3 h, and the bacteria were recovered by centrifugation at 6000× *g* for 30 min at 4 °C. Then, the supernatants were discarded, and the bacterial pellets were resuspended in sterile PBS (ratio of 30 mL PBS per 50 mL culture) and stored at −80 °C until use. To purify the cytokine-based nanoparticles, thawed bacteria were disrupted for 2 rounds at 40 KPsi (with a Constant Systems CF1 disruptor) and ice-coated with protease inhibitors (cOmplete protease inhibitor cocktail EDTA-free, Roche, Basel, Switzerland). After a new freeze–thaw cycle, the samples were incubated for 2 h with 0.01 mg/mL of lysozyme (Sigma-Aldrich, Madrid, Spain) at 37 °C and 250 rpm. A new freeze–thaw cycle was followed by the addition of 4 μL/mL of Triton X-100 and subsequent incubation for 1 h at RT in an orbital rotator shaker. At this point, a sterility control was performed by plating the sample aliquot on agar-GM17 plates and incubating them O/N at 30 °C. Further freeze–thaw cycles were carried out until no viable bacterial growth was detected. Following that, the mixture was incubated for 1 h with 0.25 μL of NP-40 per mL of sample at 4 °C in a rotatory shaker. Then, 0.6 μg/mL DNase I and 0.6 mM MgSO_4_ (Panreac, Barcelona, Spain) were added and incubated for 1 h at 37 °C and 250 rpm. The samples were centrifuged at 6000× *g* for 30 min at 4 °C. The pellets containing nanoparticles (IBs) were resuspended with 5 mL of lysis buffer (50 mM Tris-HCl, pH 8; 100 mM NaCl; 1 mM EDTA; and 0.5% (*v*/*v*) Triton X-100), frozen, and thawed again. The resultant mixture was centrifuged at 6000× *g* for 30 min at 4 °C, and the pellets were resuspended in sterile PBS and aliquoted. Finally, centrifugation at 15,000× *g* for 30 min at 4 °C was carried out, storing the IB pellets at −80 °C until use.

The IB aliquots were tested for sterility in agar-GM17 plates, incubating them O/N at 30 °C. In addition, they were quantified by Western blot using an anti-His antibody (Santa Cruz, Dallas, TX, USA) and pure GFP-(His)6 from 1000 to 250 ng, as standard. Their purity was also evaluated by performing a Coomassie blue staining assay. The outcomes were analyzed by ImageJ software to determine both protein quantity and purity.

### 2.4. Immunoassays

IPEC-J2 cells were cultured at 37 °C and 5% CO_2_ until confluence, an after trypsinization, they were seeded in 24-well plates at a density of 100,000 cells/well. The alveolar macrophages were resuspended in DMEM medium (supplemented as explained before) and centrifuged at 560× *g* for 10 min at 10 °C. Then, the pellets were resuspended in fresh DMEM medium and seeded in 24-well plates at a density of 100,000 cells/well. Prior to the immunoassay, the medium was removed, followed by an addition of 300 μL of fresh medium and the resuspended cytokine-based IB treatment in 200 μL of sterile PBS, reaching 500 μL/well. Each treatment was analyzed in sextuplicate. In both experiments, PBS, LPS, and GFP nanoparticles (IB^GFP^) were used as negative control, positive control, and format control, respectively. The cultures were incubated for 16 h at 37 °C and 5% CO_2_. Supernatants from the IPEC-J2 and alveolar macrophages were collected and kept at −80 °C, and IPEC-J2 RNA was recovered using a TRIzol^®^ (Invitrogen, Waltham, MA, USA) extraction method according to the manufacturer instructions. The experiments performed to analyze the secretion of cytokines by ELISA were carried out using nanoparticles at 10 μg cytokine/mL, whereas the experiments run for gene expression analyses were carried out using nanoparticles at 6.25 μg total protein/mL to avoid saturation in the gene expression analyses.

### 2.5. Gene Expression Analyses

The RNA was quantified using a NanoDrop^TM^ device (ThermoFisher Scientific, Waltham, MA, USA), and their integrity was analyzed by electrophoresis in 1.5% agarose gel. cDNA synthesis was performed using a PrimeScript RT reagent kit (Takara Bio Inc., Otsu, Shiga, Japan) according to the manufacturer instructions. In addition, qPCR with SYBR green (SYBR Premix Ex Taq II, Perfect Real Time, Takara Bio Inc, Otsu, Shiga, Japan) was implemented using a BioRad real-time PCR thermocycler. Briefly, an initial denaturalization was performed at 95 °C for 10 min. Next, 40 cycles of denaturalization at 95 °C for 10 s, as well as annealing and extension at 60 °C for 30 s, were performed. Finally, one cycle of 1 min at 95 °C was carried out, and the specificity of the amplified products was assessed by melting curve (61 cycles at a thermal gradient of 65 to 95 °C in 30 s). Several genes related to the inflammatory profile (*β-defensin-1* (*BD1*), *β-defensin-2* (*BD2*), *IL-6, TNF-α*) and intestinal integrity (*occludin* and *claudin-4* (*CLDN4*)) were analyzed in IPEC-J2 using ribosomal protein L4 (*RPL4*) as the housekeeping gene [18]. The primer sequences and parameters are reported in Table 1. The resulting Cp values were used to calculate the relative expression of the selected genes by relative quantification using the reference gene (housekeeping gene) and the calibrator of the control group.

**Table 1 animals-12-01075-t001:** Primers and PCR conditions (T° of annealing (°C), optimal primer concentration (μM), and PCR product (bp)) for the selected target genes. Fw: forward; Rv: reverse; bp: base pairs [18,19,20].

Target Gene	Primer Name	Sequence (5′-3′)	T° Annealing (°C)	Conc (μM)	PCR Product (bp)	Reference
Interleukin-6	IL6-Fw	CAAGGAGGTACTGGCAGAAA	60	0.25	185	
	IL6-Rv	CAGCCTCGACATTTCCCTTAT				
β-defensin 1	BD1-Fw	TGCCACAGGTGCCGATCT	60	0.25	81	
	BD1-Rv	CTGTTAGCTGCTTAAGGAATAAAGGC				
β-defensin 2	BD2-Fw	ACCTGCTTACGGGTCTTG	60	0.25	168	
	BD2-Rv	CTCTGCTGTGGCTTCTGG				
Tumor necrosis factor-α	TNFa-Fw	ATCGGCCCCCAGAAGGAAGAG	60	0.25	351	[19]
	TNFa-Rv	GATGGCAGAGAGGAGGTTGAC				
Claudin-4	CLDN4-Fw	CGCCCTCATCGTCATCTGTATC	60	0.25	121	
	CLDN4-Rv	GGCCACGATCATGGTCTTG				
Mucine-1	Muc1-Fw	GTGCCGACGAAAGAACTG	60	0.25	187	
	Muc1-Rv	TGCCAGGTTCGAGTAAGAG				
Occludin	Occludin-Fw	GCTTTGGTGGCTATGGAAGT	60	0.5	157	
	Occludin-Rv	CCAGGAAGAATCCCTTTGCT				
Ribosomal protein L4	RPL4-Fw	CAAGAGTAACTACAACCTTC	60	0.5	122	[18]
	RPL4-Rv	GAACTCTACGATGAATCTTC				
Glyceraldehyde 3- phosphate dehydrogenase	GDPH-Fw	GTCGGTTGTGGATCTGACCT	60	0.2	135	[20]
	GDPH-Rv	TCACAGGACACAACCTGGTC				
TATA-Box Binding Protein	TBP-Fw	AACAGTTCAGTAGTTATGAGCCAGA	63	0.2	153	[18]
	TBP-Rv	AGATGTTCTCAAACGCTTCG				

### 2.6. Enzyme-Linked Immunosorbent Assay (ELISA)

The supernatants of both immunoassays were used for the determination of the swine cytokines IL-6 and TNF-α secreted by the cultures under nanoparticle treatment using commercial ELISA kits (Kingfisher, London, UK) and following the manufacturer instructions. Each sample was assayed in duplicate and diluted four times when required.

### 2.7. Temperature and pH IB Stability

The cytokine nanoparticle stability was tested mimicking swine gastrointestinal conditions and temperature experienced during their possible inclusion as a feed additive in piglet concentrate. To simulate the gastrointestinal tract environment, the nanoparticles were incubated for 2 h at a pH of 4 at 37 °C, followed by 5 h at a pH of 6.5 at 37 °C. On the other hand, to simulate the temperature potentially faced during the feed production process, the IBs were incubated for 1 min at 80 °C. Then, in both assays, a Coomassie blue staining assay was performed to evaluate if the protein embedded was solubilized or was lost by degradation. For this, the samples were centrifuged at 6000× *g* for 1 min and loaded onto SDS-PAGE gel. In addition, an immunoassay using IPEC-J2 cells was conducted to evaluate if the nanoparticle immunogenicity was maintained after pH and temperature treatment.

### 2.8. In Vivo Assay

All animal experimentation procedures were approved by the Animal Ethics Committee (CEEAH) of the Universitat Autònoma de Barcelona (reference number: 9019/10548/2017) and were performed in accordance with the European Union guidelines for the care and use of animals in research (Directive 2010/63/EU).

A total of 20 piglets were selected for the study, ensuring the best litter homogeneity. The piglets were weaned around 21 days of age and were housed individually (1 animal/pen). All experimental basal diets were formulated to ensure piglet requirements. The feeding program included creep feed (from weaning to 11 days after weaning (AW)), prestarter feed (12 days AW to 27 days AW), and starter feed (from 28 days AW to 34 days AW) presented in mash form. Solid feed and water were offered *ad libitum* during the trials.

An initial phase of 11 days was conducted to acclimate the animals to the facilities, and in the following week, the animals were submitted to an operant conditioning scheme to adapt them to the selected strategy of our target administration. Specifically, a round plate with 150 g of 0.5 M sugar solution was offered every morning at 9:00 am until the animal finished its content

In the trial, 2 treatments were included (*n* = 10 animals each): control (animals received a 0.5 M sugar solution) and treatment (animals received IL-1β nanoparticles in 0.5 M sugar solution.) The immunostimulant treatment based on IL-1β nanoparticles (20 µg total protein/kg of BW) was applied for 7 days in a round plate following the trained routine previously described. Twenty-four hours after the last administration (day 27), half of the animals of each treatment (*n* = 5) were blood-sampled and euthanized for tissue sampling. The rest of the animals were similarly sampled (for blood and tissues) 7 d after (day 34) the last administration. This 7-day sampling delay was decided in order to evaluate the effect of nanoparticle slow-release over time in the piglet immune profiles. The concentrations of inflammation-related proteins IL-8, IL-6, TNF-α, and IL-10 were quantified by ELISA in the blood.

### 2.9. Statistical Analysis

The immunoassays were performed in sextuplicate and cytokine stability experiments in triplicate, with all represented as the means of nontransformed data ± standard error of the mean (SEM).

All data were tested for normality using JMP software (SAS Institute Inc., Cary, NC, USA) The data were log-transformed when needed and analyzed using the MIXED procedure of SAS (9.4, SAS Institute, Cary, NC, USA). The model included treatment, day of tissue sampling, and their interaction as the main effect. Differences were declared significant at *p* < 0.05, and trends were discussed at 0.05 ≤ *p* ≤ 0.10.

## 3. Results

### 3.1. Production and Characterization of Cytokine-Based Nanoparticles

Four cytokines involved in the inflammatory response (IL-1β, IL-6, IL-8, and TNF-α) were produced recombinantly in *Lactococcus lactis* as protein nanoparticles (IBs). GFP was also produced as a nonimmune-related control nanoparticle. The protein yield of the nanoparticles and the estimated cytokine content are depicted in Table 2. The IL-8 cytokine was the best-produced nanoparticle, whereas TNF-α was purified at such low levels that it was not quantifiable by Western blot. IL-1β, IL-6, and the GFP control were produced at moderate yields ranging between 0.5 and 1.67 mg/L of culture. In all cases, the cytokines comprised from 11 to 34% of the nanoparticle composition, indicating that other proteins from the *L. lactis* host were also present.

### 3.2. Immunostimulation of Swine Intestinal Cells and Macrophages

The immunostimulation potential of the nanoparticles was tested on porcine intestinal cells and alveolar swine macrophages at a final concentration of 10 μg cytokine/mL by monitoring the induction of TNF-α and IL-6 secretion (Table 3). The highest stimulation of the alveolar macrophages was caused by IL-8- and IL-1β-containing nanoparticles, boosting the secretion of TNF-α and IL-6, respectively. The positive control used was LPS, and it performed equally to the IL-8-based nanoparticles (Table 3). The IL-6 and TNF-α cytokine-based nanoparticles did not increase the secretion of inflammation markers compared to basal levels of PBS-treated cells or the negative control GFP nanoparticles (Table 3). The GFP-based nanoparticles slightly increased the basal levels of IL-6 secretion compared to PBS control (Table 3).

On the other hand, IPEC-J2 intestinal cells showed a less reactive pattern than macrophages, and only IL-6 secretion was detected after stimulation with nanoparticles containing IL-1β (Table 3). Neither LPS at 10 μg/mL nor other cytokines induced any inflammation in the epithelial cells, although TNF-α nanoparticles slightly boosted epithelial TNF-α secretion (Table 3). In order to increase the sensitivity and have an idea of the effect of nanoparticles on intestinal epithelial cells, the gene expressions of several genes involved in innate immunity were assessed (Figure 1). In this assay, the treatments were applied based on the total protein content of the nanoparticle (6.25 μg total protein/mL) rather than the cytokine concentration partially because the cytokine embedded in the nanoparticle could trigger an inflammatory response. The results confirmed that IL-1β nanoparticles boosted an inflammatory response in epithelia, increasing the gene expression of *TNF-α* (Figure 1). Moreover, the gene expression profile also confirmed that TNF-α-based nanoparticles upregulated *TNF-α* and *CLDN4* genes, whereas IL6 nanoparticles increased the expression of the *BD2* and *CLDN4* genes (Figure 1). Herein, PBS did not show any effect on gene expression, while GFP induced *CLDN4* expression.

### 3.3. Temperature and pH Stability of Cytokine Nanoparticles

The nanoparticles containing either IL-1β, IL-8, IL-6, TNF-α, or GFP were incubated at high temperatures and a low pH to determine their stability. The fluctuation of protein content was determined in all cases, except for TNF-α, which was not possible to quantify by Coomassie assay (Figure 2). In all scenarios, the protein content was maintained, and we did not register significant losses towards either the soluble fraction or degradation. The immunostimulation performance was assessed by *TNF-α* expression in the epithelial. In all cases, the immunogenic activity was maintained, except for TNF-α nanoparticles, which lost activity after the temperature challenge (Figure 3).

### 3.4. Swine In Vivo Experiments

Since the IL-1β nanoparticles showed adequate production yields (Table 2), fine modulation of inflammatory responses (Table 3, Figure 1), and intrinsic resistance to in vitro simulated gastrointestinal (GIT) conditions, they were chosen to be tested in vivo in piglets. The results showed that, 24 h after the last IL-1β nanoparticle administration (Table 4), none of the analyzed cytokines in the blood showed significant differences from the control. However, TNF-α tended to increase at 7 d post-administration (day 34) (Table 4, *p* = 0.0755) of the IL-1β treatment compared to control piglets.

In samples of intestinal tissue, the immunostimulatory effect was assessed by the gene expression of the extracted RNA from tissue explants after 24 h (day 27) and 7 d (day 34) of the administration of IL-1β nanoparticles (Table 5). No significant changes were observed in the ileum or jejunum for *TNF-α*, *IL-6*, *BD1*, *BD2*, *Muc1*, *CLDN4*, or *Occludin* genes for both evaluated sampling times.

## 4. Discussion

Torrealba et al. [13] showed that IBs, protein nanoparticles formed during recombinant protein production, presented excellent immunomodulatory properties able to protect fish against otherwise lethal bacterial challenges. Likely, the composition and structured organization of IB components (protein peptidoglycan, DNA, and RNA) make these protein biomaterials excellent immunogens [13]. Moreover, the authors showed that, when the recombinant protein produced was a cytokine such as TNF-α or CCL4, the nanoparticles were able to interact with relevant immune cells and tissues both when intraperitoneally injected or orally administrated and provided better protection levels compared to similar nanoparticles that included proteins without any specific immune function [21]. These conclusions pushed us to test this concept in swine production as an alternative approach to increase piglet resilience during stressful periods and reduce associated antibiotic use.

In the present study, *L. lactis* was the recombinant platform used to produce the cytokine-based nanoparticles since it is considered a generally recognized as safe (GRAS) system, and it would facilitate potential implementation as a feed additive for animal production [22]. Nanoparticles based on IL-1β were the only ones that stimulated the immune response both in macrophages and intestinal epithelia by increasing IL-6 secretion above the levels shown by control cells treated with PBS or GFP nanoparticles. The IL-8 nanoparticles also stimulated alveolar macrophages by increasing TNF-α secretion but did not produce any effect on IPEC-J2 cells. LPS added at the same concentration as nanoparticles (10 μg/mL) increased TNF-α in macrophages but did not induce innate immunity in IPEC-J2 cells. This indicated that, as expected, the reactivity of macrophages was much greater than intestinal cells, although the latter were still able to respond to IL-1β nanoparticle stimulus by secreting the pro-inflammatory cytokine IL-6. The gene expression in IPEC-J2 cells was evaluated, as this is considered to have a higher sensitivity than ELISA tests. We selected genes covering not only pro-inflammatory cytokines, such as *IL-6* and *TNF-α*, but also host defense peptides (HDPs) such as *β-defensin* 1 (*BD1*) and 2 (*BD2*), which play an important role in innate immunity fighting against pathogens. Finally, two genes involved in the formation of tight junctions, *Occludin* and *CLDN4*, were selected since their increase prevents the entrance of pathogens inside the cell [23]. We indeed found an upregulation of the *TNF-*α gene by IL-1β nanoparticles and an increase of *TNF-α* and *CLDN4* genes in cells treated with TNF-α nanoparticles. It was unexpected that *IL-6* expression was not detected since it was well-detected by ELISA. However, this was possible because the sampling time for the IL-6 expression analyses was not the optimum. However, in vitro we found very interesting activity of the TNF-α nanoparticles. It is important to state that the production yield of this nanoparticle was extremely low, which makes it difficult to consider it as a candidate for further exploration. The nanoparticles based on IL-6 also stimulated the expression of two genes in IPEC-J2, *CLDN4* and *BD2*, and since their yield was acceptable, they can be considered possible candidates for future in vivo experiments. However, although the IL-6 and IL-8 nanoparticles were able to stimulate macrophages, the IL-1β nanoparticles induced both macrophages and intestinal cells, which makes them more attractive. Lastly, we also found that GFP nanoparticles induced a greater expression of *CLDN4* than the PBS control. In a previous study, Torrealba et al. [21] also found unspecific immunogenicity responses in vitro using nanoparticles with control proteins such as iRFP, but in the in vivo studies, they demonstrated that the effect was better using nanoparticles containing immune-relevant proteins, such as cytokines [21]. It should be noted that all cytokine-based nanoparticles produced herein had a low purity of recombinant protein, indicating that the immunogenicity was probably caused by a combination of several components embedded in the IBs. The stability experiments demonstrated that the cytokine nanoparticles were highly stable, regarding both activity and protein content, at low pH and physiological temperature (37 °C). Thus, they could resist gastrointestinal conditions. Also, they supported high-temperature conditions usually used in animal feed preparation, which makes them suitable for possible applications as a feed additive.

Considering all the in vitro results, the IL-1β-based nanoparticles were chosen as a first proof-of-concept in a small number of piglets to assess if the immunostimulation was translated at an animal level. This was a first assessment of their potential using a single dose without any specific challenge since it will be tested in further experiments. Although the use of cytokines as immunostimulants has been broadly proposed in animals, no clear conclusions can be obtained regarding optimum concentrations. Many studies have been performed by delivering expression vectors of cytokines, where the exact produced amount has been difficult to determine [24]. On the other hand, earlier studies have reported the amount of cytokine used in activity units as a parameter without enough traceability. Among the available information, IL-12 has been used in several clinical trials with mice for the treatment of cancer at concentrations ranging from 0.5 to 1 μg/5–6-week-old mouse (approx. 0.05 mg/kg) over 6 consecutive days and has resulted in 100% mortality because of toxic effects. However, similar doses of IL-12 given i.n. were relatively well-tolerated [25]. Torrealba et al. demonstrated that a unique dose of 7.49 ± 0.97 mg of TNF-α nanoparticles/kg had clear prophylactic potential in vivo when protecting zebrafish from a lethal infection of *Pseudomonas aeruginosa* [21]. Finally, it was demonstrated that the administration of 1 mg/kg of IFN-γ 6 h before *Salmonella* infection and continuing for 5 days had a clear beneficial effect on calves [26]. The dose applied herein was limited to the production that could be achieved at lab-scale and corresponded to a daily administration of 20 μg nanoparticles/kg of body weight for 1 week. This dose was considered reasonable since the real amount of cytokine was 16% (3 μg cytokine/kg). Moreover, we considered that a previous experiment with LPS at 2 µg/kg induced immunostimulation in piglets [27]. However, the effects detected in the intestinal explants of piglets by gene expression were not significant for a broad range of genes involved in mucosal immune response, such as cytokines, mucins, tight junction proteins, or HDPs (Table 5). However, interestingly, the TNF-α concentration in the blood of animals treated with IL-1β nanoparticles tended to be greater at 7 d post-administration compared to control piglets. Since the number of animals was limited, it was difficult to obtain a significant effect, but the blood TNF-α concentrations tended to be 100 times greater than in controls (Table 4).

Other studies exploring alternative immunostimulants based on probiotics such as *Bacillus subtilis* and lactic acid bacteria have observed changes in gene expression at the intestinal level, including a clear increase in *IL-6* gene expression [11]. In these cases, the administration of probiotics lasted for around 3 weeks, which could suppose a relevant difference in conjunction with the selected strategy to trigger immunostimulation [11]. Another approach using nonviable microorganisms was tested. Zhong et al. demonstrated that the intestinal mucosal and systemic immunity of early-weaned piglets were reinforced by heat-killed *Mycobacterium phlei* but not by antibiotics [28]. However, there was also another study exploring shorter treatments of 11 d using phytobiotics. For example, 10 mg/kg of *Capsicum Oleoresin*, *Garlic Botanical*, or *Turmeric Oleoresin* upregulated the expression of genes related to immune response in supplemented animals compared to the control [29].

Previous works have also explored the effect of immunostimulants on systemic immunity. However, in most cases, the focus has been on the concentration of immunoglobulins, which was not assessed in our case [30]. For example, the immune active protein lactoferrin was studied in weaning piglets, increasing PHA-stimulated lymphocyte proliferation, serum (IgG by 16%, IgA by 17%, and IL-2 by 14% (*p* < 0.05)), and serum iron values (23%, *p* < 0.01), as well as decreasing the diarrhea ratio (*p* < 0.05) relative to control on day 30 [31]. However, in the lactoferrin study, a much greater dose of 1 g/kg was administered for days 15 and 30.

## 5. Conclusions

Immunostimulation is a compelling strategy to prevent nondesirable infections. This approach is underpinned on a proper application of adequate animal production timescale. The preliminary outcomes demonstrated that our cytokine-based nanoparticles (specially IL-1β and IL-6) were able to immunostimulate in vitro swine intestinal cells and macrophages, even after a temperature and pH challenge. Going a step further, the selected cytokine for in vivo assays was IL-1β, but although they showed a good and stable in vitro performance, IL-1β nanoparticles did not elicit significant effects in vivo. However, a tendency was observed to have an immune stimulatory effect at a systemic level, which could increase animal resilience to infections. It is possible that greater doses and longer treatment durations may be needed to detect a pronounced effect in the intestinal mucosa, along with a comprehensive evaluation of the optimal treatment application timeframe.

## Figures and Tables

**Figure 1 animals-12-01075-f001:**
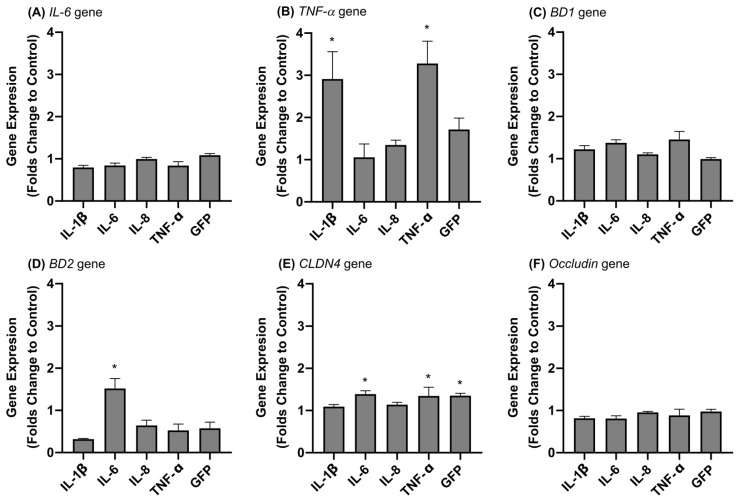
Analysis of gene expressions of (**A**) *IL-6,* (**B**) *TNF-**α,* (**C**) *BD1,* (**D**) *BD2,* (**E**) *CLDN4*, and (**F**) *Occludin,* in folds compared to negative control in the IPEC-J2 cell line. Grey bars indicate treatment with 6.25 μg total protein/mL. GFP IBs and PBS were used as format control and negative inflammatory control, respectively. Error bars indicate the standard error of the mean (SEM). Asterisks show statistically significant differences in expression folds between PBS and treatments. (**A**) *p =* 0.030; (**B**) *p* = 0.0004; (**C**) *p* = 0.0108; (**D**) *p* = 0.0001; (**E**) *p* = 0.0006; (**F**) *p* = 0.5059.

**Figure 2 animals-12-01075-f002:**
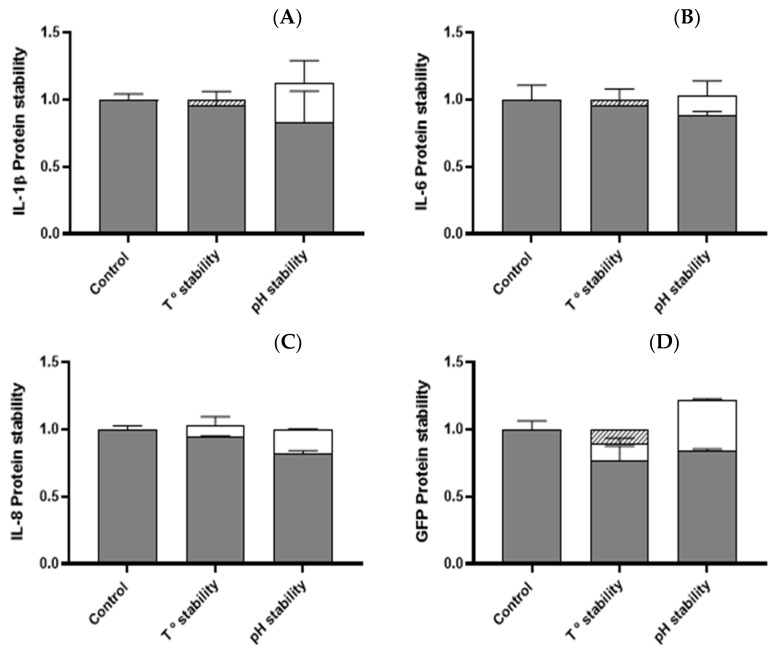
Protein distribution between soluble (white), insoluble (grey), or degraded fractions (striped) after temperature and pH challenge in (**A**) IL1β-, (**B**) IL6-, (**C**) IL8-, and (**D**) GFP-based IBs.

**Figure 3 animals-12-01075-f003:**
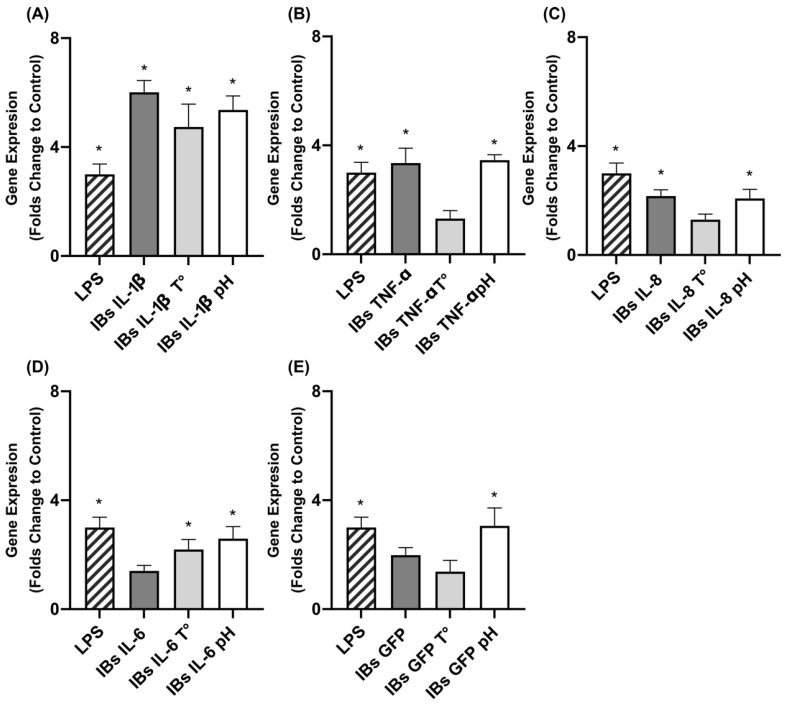
TNF-α gene expression (in folds compared to negative control) after temperature and pH stability assay in IPEC-J2 cells treated by (**A**) IL-1β-, (**B**) TNF-α-, (**C**) IL-8-, (**D**) IL-6-, and (**E**) GFP-based nanoparticles. The bars of mean standard deviation and SEM of nontransformed data are represented. Asterisks display statistically significant differences in expression folds between PBS and treatments. (**A**) *p* = 0.0001; (**B**) *p* = 0.0001; (**C**) *p* = 0.0003; (**D**) *p* = 0.0002; (**E**) *p* = 0.0019.

**Table 2 animals-12-01075-t002:** Cytokine IB yields (mg/L culture) and recombinant protein content (%) of each cytokine nanoparticle produced in *L. lactis*; n.d: nondetected.

Cytokine	Yield (mg/L) ^a^	Recombinant Protein Content (%)
Interleukin-1β	0.51 ± 0.20	16.19 ± 0.06
Interleukin-6	1.19 ± 0.12	34.40 ± 0.04
Interleukin-8	25.69 ± 3.49	23.39 ± 6.82
Tumor necrosis factor-α	n.d	n.d
GFP	1.67 ± 0.15	11.01 ± 1.39

^a^ yield obtained after IB purification process.

**Table 3 animals-12-01075-t003:** Inflammatory response of alveolar macrophages and intestinal epithelial cell line of swine (IPEC-J2). The secretion of IL-6 and TNF-α (ng/mL) was evaluated by ELISA after treatment with 10 μg/mL cytokine-based IBs containing IL-1β, IL-6, IL8, or TNF-α. GFP IBs were used as a format control. LPS (10 μg/mL) and PBS were employed as positive inflammatory control and negative control, respectively. Means and standard error of the mean (SEM) from nontransformed data are represented. Asterisks depict significant differences from PBS control; *p* < 0.0001; n.d: nondetected.

Tissue	IB Treatment (10 μg/mL)	IL-6 Secretion (ng/mL)	TNF-α Secretion (ng/mL)
Alveolar macrophages	IL-1β	8.868 ± 0.182 *	n.d
	IL-6	n.d	n.d
	IL-8	0.031 ± 0	4.622 ± 0.109 *
	TNF-α	0.381 ± 0 *	1.206 ± 0 *
	GFP	2.235 ± 0.016 *	0.796 ± 0.091 *
	LPS	0.067 ± 0.010 *	5.277 ± 0.062 *
	PBS	0.036 ± 0.018	2.214 ± 0.061
Intestinal Epithelial cells (IPEC-J2)	IL-1β	21.125 ± 4.598 *	n.d
	IL-6	n.d	n.d
	IL-8	n.d	n.d
	TNF-α	0.300 ± 0.006	2.646 ± 0.055
	GFP	7.006 ± 0.403 *	n.d
	LPS	0.018 ± 0.004	n.d
	PBS	0.015 ± 0.003	n.d

**Table 4 animals-12-01075-t004:** Cytokine determination (ng/mL) in serum samples from in vivo swine treatments with IL-1β nanoparticles and in the control treatment. The mean of each treatment and SEM are indicated. Highlighted results indicate a tendency. T: treatment; D: day.

	Day-27		Day-34		SEM	*p* Value		
Cytokine	Control	Treatment	Control	Treatment		T	D	TxD
IL-8	0.066	0.097	0.089	0.251	0.080	0.116	0.476	0.112
IL-6	0.435	0.620	0.511	1.324	0.286	0.136	0.105	0.328
TNF-α	1.188	0.522	**0.570**	**54.106**	16.370	0.122	0.120	0.076
IL-10	0.350	0.680	0.300	3.202	1.569	0.854	0.697	0.900

**Table 5 animals-12-01075-t005:** Cytokine gene expression analysis of ileum (A) and jejunum (B) after IL-1β-based IB treatment and in the control treatment. The mean of each treatment and SEM are indicated. *p*-value < 0.05 indicates statistical differences. W: week.

		Day-27		Day-34		SEM	*p* Value		
Tissue	Gene	Control	Treatment	Control	Treatment		T	D	TxD
Ileum	TNF-α	2.8 × 10^−9^	0.265	0.244	0.423	0.527	0.679	0.708	0.936
(A)	IL6	1.009	0.953	1.020	1.045	0.144	0.917	0.724	0.784
	BD1	1.416	1.393	3.175	2.898	1.279	0.679	0.708	0.936
	BD2	1.092	0.873	1.594	1.165	0.251	0.215	0.133	0.680
	Muc1	1.139	0.677	1.064	0.970	0.173	0.129	0.539	0.304
	CLDN4	1.023	0.832	1.219	0.998	0.105	0.068	0.104	0.890
Jejunum	TNF-α	1.030	1.153	0.948	0.807	0.113	0.938	0.077	0.261
(B)	IL6	1.039	0.964	0.877	0.783	0.144	0.566	0.251	0.950
	BD1	1.707	1.761	1.990	5.290	2.130	0.917	0.362	0.321
	BD2	1.118	1.555	1.550	3.406	0.922	0.225	0.155	0.497
	Muc1	1.005	1.361	1.232	1.470	0.187	0.131	0.382	0.756
	CLDN4	1.031	0.870	0.935	0.956	0.105	0.515	0.959	0.398
	Occludin	1.017	1.001	1.116	1.085	0.106	0.827	0.401	0.944

## Data Availability

The data presented in this study are available on request from the corresponding author.

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
