# Peer review of "Potential of Oral Nanoparticles Containing Cytokines as Intestinal Mucosal Immunostimulants in Pigs: A Pilot Study"

_animals, 2022, doi:10.3390/ani12091075_

Round 1

Reviewer 1 Report

The manuscript animals-1642496 entitled 'Potential of oral Nanoparticles containing cytokines as Intestinal Mucosal Immunostimulants in Pigs: a pilot study' falls with its subject within the scope of Animals, but requires revision before re-submitting. The list of proposed changes is given below:

  1. The authors reported the potential of oral nanoparticles containing cytokines as intestinal mucosal immunostimulants in pigs, which was very interesting but lack of safety evaluation. Authors should analyze and assesse the application risk of oral nanoparticles containing cytokines. The traditional immunostimulants, for example some compounds based on flavonoids, essential oils, probiotics, or prebiotics, have been deeply explored for livestock applications, and have greater security. However, inflammatory cytokines, including IL1β, IL8 and TNFα, are typical pro-inflammatory factors, closely linking with many diseases. These molecules may exert toxic effects toward mucosal cells.

  1. This study has a fewer number of test animals in the vivo experiment (N =5 /treatment). Meanwhile, feed intake, body weight and the health status of animals should be recorded, especially whether the piglets suffered diarrhea, dystrophia and slow growth.

  1. Authors believed that the most important function of immunostimulants was the immune stimulatory effects at systemic level, which could increase animal resilience to infections. But in this current study, authors did not analyze the responses of pigs with the nanoparticles when occurring pathogen infection. This is very regrettable.

  1. Which kind of animals is the prime target group of this immunostimulant? Piglets? However, it is not easy to maintain the balance of intestinal micro-ecological in piglets, especially during the weaning period. In this case, I want to know If the use of immunostimulant intensify the stress response.

Author Response

The manuscript animals-1642496 entitled 'Potential of oral Nanoparticles containing cytokines as Intestinal Mucosal Immunostimulants in Pigs: a pilot study' falls with its subject within the scope of Animals, but requires revision before re-submitting. The list of proposed changes is given below:

 The authors reported the potential of oral nanoparticles containing cytokines as intestinal mucosal immunostimulants in pigs, which was very interesting but lack of safety evaluation. Authors should analyze and assesse the application risk of oral nanoparticles containing cytokines. The traditional immunostimulants, for example some compounds based on flavonoids, essential oils, probiotics, or prebiotics, have been deeply explored for livestock applications, and have greater security. However, inflammatory cytokines, including IL1β, IL8 and TNFα, are typical pro-inflammatory factors, closely linking with many diseases. These molecules may exert toxic effects toward mucosal cells.

AU: We agree with reviewer on the necessity to test the safety of these immunoestimulants. However, we would like to point out that the aim of this paper is a first exploration of nanoparticles containing cytokines towards their future use as immunoestimulant in porcine applications. In fact, the idea is to get an overview of their potential and evaluate how to continue their study. We have added some sentences in in the manuscript to clarify this point. Besides, in the future, we plan to perform safety experiments after testing several doses with more animals and using a model that includes an infection challenge.

Although we plan to perform safety experiments with our nanoparticles, we would like to let the reviewer know that previous studies using cytokine-containing nanoparticles for aquaculture applications have demonstrated that there is no toxicity on zebrafish liver cells (Torrealba et al. Nanostructured recombinant cytokines: A highly stable alternative to short-lived prophylactics.Biomaterials. 2016 Nov;107:102-14). Moreover, the format of nanoparticles based on inclusion bodies was tested in vivo using mice and no toxic effects were described (Vazquez et al. Functional inclusion bodies produced in bacteria as naturally occurring nanopills for advanced cell therapies. Adv Mater. 2012 Apr 3;24(13):1742-7).

This study has a fewer number of test animals in the vivo experiment (N =5 /treatment). Meanwhile, feed intake, body weight and the health status of animals should be recorded, especially whether the piglets suffered diarrhea, dystrophia and slow growth.

AU: As stated in the paper the in vivo study is just a pilot study to roughly explore the effects of nanoparticles. Most of the experiments of the paper are focused on in vitro exploration with the idea to do a more accurate in vivo study with more animals. From this first study we have learnt that the dose used did not induce a clear immunostimulatory effect, so in future trials we could increase the dose and we could also set up an administration methodology. Moreover, it would be probably necessary to do a challenge since we have not observed any signs of diarrhea. We have not included the weight of the animals because the number is too low to have relevant results but we recorded the weight and also health status and no differences were observed (we have included this information in the text of the manuscript).

Authors believed that the most important function of immunostimulants was the immune stimulatory effects at systemic level, which could increase animal resilience to infections. But in this current study, authors did not analyze the responses of pigs with the nanoparticles when occurring pathogen infection. This is very regrettable.

AU: We completely agree with the reviewer that future idea is to test if immunoestimulants could increase animal resilience. However, this is a preliminary study focused on the in vitro evaluation of the potential of several cytokine nanoparticles and to perform a first assessment by using a low number of animals. We are working on the design of further studies setting up a challenge with intestinal pathogens and using a high number of animals per group.

Which kind of animals is the prime target group of this immunostimulant? Piglets? However, it is not easy to maintain the balance of intestinal micro-ecological in piglets, especially during the weaning period. In this case, I want to know If the use of immunostimulant intensify the stress response.

AU: Several applications could be used for an immunostimulant in animal production involving critical periods of infections (piglets or calves at weaning, before transport, in mothers after delivery to increase immune components in colostrum,...). We have used piglets as a first model to explore this hypothesis but we agree with the reviewer that to demonstrate if there is an increase resilience to diarrhea for example a real challenge with bigger number of animals must be tested under. As stated before, this will be the focus of further studies. Before testing their performance in a challenge

it is important to set up a dose in which the effect is not too heavy to avoid increasing the stress suffered for the animal and also to adjust the administration time to have slightly activated the immune system before the challenge occurs. In conclusion, we totally agree with the reviewer that we need a deep exploration of this use but this present study is just the tip of the iceberg of cytokine nanoparticles.

Reviewer 2 Report

In this manuscript, the authors present a pilot study to evaluate the potential of inclusion bodies (IBs) from recombinant cytokines as immunomodulators of intestine epithelial cells in vitro and also in vivo in piglets.

Arising antimicrobial resistance and the increasing need for alternatives to antibiotics are major concerns and it is valuable to develop new avenues.

The materials and methods section needs to be improved to fit the results better. For example, the immunoassay section must be improved to indicate the quantity of IBs that was used in the different experiments. Moreover, the results section indicate that different quantities were used (the total protein content or specific cytokine content). This must be clearly stated.

The M&M section must also hint that following in vitro results, the Il-1β IBs were selected for the in vivo experiments.

Please provide more information on the protein quantification. My understanding is that it was determined by gel and westernblot analysis. Was there a standard included in the gel? The wersternblot? Quantification is a critical point of the whole study and more information is needed to validate quality of the results obtained. Give also more information about that in the footnote of table 2.

Please provide more information about the “folds” used for qPCR results, in the M&M section and also in the footnotes of the figures 1 and 3.

I’m not sure I understand figure 2. What are the soluble, insoluble and lost fractions ? Does the lost fraction represent the degradation ? Section 2.7 of the M&M and the legend of the figure 2 must be improved.

Figure 4: what is the units?

Figure 5: what is the unit?

Why the non-immune related control IBs (GFP IBs) were not used as a negative control in the in vivo experiment. Is it not a better negative control ?

L348: “the bars of the mean” ?

L354: suggestion: in vitro simulated gastrointestinal (GIT) conditions.

Author Response

In this manuscript, the authors present a pilot study to evaluate the potential of inclusion bodies (IBs) from recombinant cytokines as immunomodulators of intestine epithelial cells in vitro and also in vivo in piglets.

Arising antimicrobial resistance and the increasing need for alternatives to antibiotics are major concerns and it is valuable to develop new avenues.

The materials and methods section needs to be improved to fit the results better. For example, the immunoassay section must be improved to indicate the quantity of IBs that was used in the different experiments. Moreover, the results section indicate that different quantities were used (the total protein content or specific cytokine content). This must be clearly stated.

AU: We have modified the M&M and Results section according to reviewer comments

The M&M section must also hint that following in vitro results, the Il-1β IBs were selected for the in vivo experiments.

AU: The selection of Il-1β IBs is part of the results and we think that this information should not be included in M&M but in the results section, as it is.

Please provide more information on the protein quantification. My understanding is that it was determined by gel and westernblot analysis. Was there a standard included in the gel? The wersternblot? Quantification is a critical point of the whole study and more information is needed to validate quality of the results obtained. Give also more information about that in the footnote of table 2.

AU: According to the reviewer suggestion, we have included this information in the M&M and the footnote.

Please provide more information about the “folds” used for qPCR results, in the M&M section and also in the footnotes of the figures 1 and 3.

AU: We have included this information in the M&M and footnotes

I’m not sure I understand figure 2. What are the soluble, insoluble and lost fractions ? Does the lost fraction represent the degradation ? Section 2.7 of the M&M and the legend of the figure 2 must be improved.

AU: We have improved this explanation either in the M&M and in the legend of figure 2.

Figure 4: what is the units?

AU: We have included this information in the legend

Figure 5: what is the unit?

AU: We have included this information in the legend

Why the non-immune related control IBs (GFP IBs) were not used as a negative control in the in vivo experiment. Is it not a better negative control ?

AU: This is a very interesting point we considered carefully. However, we discard to include the GFP IBs as a control because of low number of animals and we consider a proper negative control the saline treated animals. The reason behind this decision was that we consider necessary to have a negative control without any kind of immunoestimulation and it was not possible using GFP IBs since they could include some traces from the production process that could induce a kind of immunoestimulation to the animals as demonstrated in previous studies with fish (Torrealba et al. Nanostructured recombinant cytokines: A highly stable alternative to short-lived prophylactics.Biomaterials. 2016 Nov;107:102-14; Torrealba et al. Complex Particulate Biomaterials as Immunostimulant-Delivery Platforms. 2016 Oct 7;11(10):e0164073).

L348: “the bars of the mean” ?

AU: Thanks for the comment. It was a mistake. We have changed it.

L354: suggestion: in vitro simulated gastrointestinal (GIT) conditions.

AU: Changed

Round 2

Reviewer 1 Report

The authors have made some improvement in the manuscript. I therefore recommend this paper to be publicated after revised. As a suggestion, authors should add some discussions about the concentrations of inflammatory factors and their effects on immune activation. The low dose cytokines can improve the immune sensitivity of the body, but the high dose cytokines may bring some potential risk, like intestinal immune imbalance. For these oral nanoparticles containing cytokines (IL1β, IL8, TNFα … …), it is essential to know the exact quantity of the drug. 

Author Response

The authors have made some improvement in the manuscript. I therefore recommend this paper to be publicated after revised. As a suggestion, authors should add some discussions about the concentrations of inflammatory factors and their effects on immune activation. The low dose cytokines can improve the immune sensitivity of the body, but the high dose cytokines may bring some potential risk, like intestinal immune imbalance. For these oral nanoparticles containing cytokines (IL1β, IL8, TNFα … …), it is essential to know the exact quantity of the drug. 

AU answer: Thanks for your comments. We have added all theses information in the discussion as suggested.

Reviewer 2 Report

Thanks you for providing this modified version of your manuscript. I have no further comments.

Author Response

Thanks very much for your comments and help at improving the manuscript.